# Plant Polyphenols Inhibit Functional Amyloid and Biofilm Formation in *Pseudomonas* Strains by Directing Monomers to Off-Pathway Oligomers

**DOI:** 10.3390/biom9110659

**Published:** 2019-10-26

**Authors:** Zahra Najarzadeh, Hossein Mohammad-Beigi, Jannik Nedergaard Pedersen, Gunna Christiansen, Thorbjørn Vincent Sønderby, Seyed Abbas Shojaosadati, Dina Morshedi, Kristian Strømgaard, Georg Meisl, Duncan Sutherland, Jan Skov Pedersen, Daniel E. Otzen

**Affiliations:** 1Biotechnology Group, Faculty of Chemical Engineering, Tarbiat Modares University, P.O. Box 14115-143, Tehran, Iran; zn@inano.au.dk; 2Interdisciplinary Nanoscience Centre (iNANO), Aarhus University, 8000 Aarhus C, Denmark; beigi@inano.au.dk (H.M.-B.); jannik@inano.au.dk (J.N.P.); thvs@inano.au.dk (T.V.S.); duncan@inano.au.dk (D.S.); 3Department of Biomedicine-Medical Microbiology and Immunology, Aarhus University, 8000 Aarhus C, Denmark; gunna@biomed.au.dk; 4Department of Industrial and Environmental Biotechnology, National Institute of Genetic Engineering and Biotechnology, P.O. Box: 1417863171, Tehran, Iran; morshedidina@yahoo.com; 5Department of Drug Design and Pharmacology, University of Copenhagen, 2100 Copenhagen Ø, Denmark; kristian.stromgaard@sund.ku.dk; 6Department of Chemistry, University of Cambridge, Cambridge CB2 1EW, UK; gm373@cam.ac.uk; 7Department of Chemistry, Aarhus University, 8000 Aarhus C, Denmark

**Keywords:** bacterial amyloid, FapC protein, extracellular matrix, aggregation inhibitor, peptide array

## Abstract

Self-assembly of proteins to β-sheet rich amyloid fibrils is commonly observed in various neurodegenerative diseases. However, amyloid also occurs in the extracellular matrix of bacterial biofilm, which protects bacteria from environmental stress and antibiotics. Many *Pseudomonas* strains produce functional amyloid where the main component is the highly fibrillation-prone protein FapC. FapC fibrillation may be inhibited by small molecules such as plant polyphenols, which are already known to inhibit formation of pathogenic amyloid, but the mechanism and biological impact of inhibition is unclear. Here, we elucidate how polyphenols modify the self-assembly of functional amyloid, with particular focus on epigallocatechin gallate (EGCG), penta-*O*-galloyl-β-d-glucose (PGG), baicalein, oleuropein, and procyanidin B2. We find EGCG and PGG to be the best inhibitors. These compounds inhibit amyloid formation by redirecting the aggregation of FapC monomers into oligomeric species, which according to small-angle X-ray scattering (SAXS) measurements organize into core-shell complexes of short axis diameters 25–26 nm consisting of ~7 monomers. Using peptide arrays, we identify EGCG-binding sites in FapC’s linker regions, C and N-terminal parts, and high amyloidogenic sequences located in the R2 and R3 repeats. We correlate our biophysical observations to biological impact by demonstrating that the extent of amyloid inhibition by the different inhibitors correlated with their ability to reduce biofilm, highlighting the potential of anti-amyloid polyphenols as therapeutic agents against biofilm infections.

## 1. Introduction

Functional amyloids are formed from proteins secreted by microorganisms which self-assemble extracellularly to β-sheet-rich fibrillar aggregates [1], typically in the form of long, needle-like structures. Unlike misfolded proteins involved in amyloid-related diseases like Parkinson’s and Alzheimer’s [2,3], expression and release of these functional amyloidogenic proteins is meticulously controlled in biofilm formation and leads to useful biological properties such as coating of endospores, stabilization, and increased antibiotic resistance [4,5]. Functional amyloids are particularly widespread in bacteria, e.g., curli in *Escherichia coli*, TasA in *Bacillus subtilis*, phenol-soluble modulins in *Staphylococcus aureus*, and Fap in *Pseudomonas* strains [6,7]. They serve a multitude of purposes in these bacteria [7,8]. In the case of the opportunistic pathogen *Pseudomonas aeruginosa*, associated with chronic airway infections of cystic fibrosis patients [9,10], amyloid in the extracellular matrix mechanically strengthens bacterial biofilm [11,12]. This increases bacterial resistance to antibiotics and environmental stress [13] and underpins chronic wound infections. Unlike pathological amyloid in e.g., neurodegenerative diseases, production and secretion of functional amyloid is carefully regulated by a number of helper proteins. Thus, the major component of this amyloid is the protein FapC, produced by the *fap* operon (*Fap A-F*) which also codes for other proteins which facilitate extracellular secretion of FapC. Of these, FapF forms a trimeric porin in the outer membrane [14] with an extended coiled-coil domain that may facilitate transport across the periplasm [15]. FapB, which is found at low levels in mature fibrils [16] and is believed to act as a nucleator to facilitate rapid FapC fibrillation. Control of functional amyloid formation is also seen at the level of the protein sequence. FapC from different *Pseudomonas* species share three imperfect repeats (R1–R3) separated by two linkers of variable size [4]. In *Pseudomonas sp UK4*, FapC consists of 250 residues, including a 24-residue signal sequence. Stepwise removal of the different repeats, particularly R3, destabilizes the ensuing FapC fibrils (L.F.B. Christensen and D.E.O., unpublished results) and increases the tendency of the fibrils to fragment during fibrillation [17].

Polyphenolic compounds constitute a large group of small molecule metabolites with antioxidant properties and a range of promising disease-fighting properties [18], not the least against protein aggregation [19,20]. Plants and fruits are natural sources of polyphenols such as epigallocatechin gallate (EGCG, found in green tea), the gallotannin penta-*O*-galloyl-β-d-glucose (PGG, pomegranates, sumac, mango seeds), baicalein (root of the flowering plant *Scutellaria baicalensis*), oleuropein (green olive leaves, skin, and flesh) and procyanidin B2 (apple, grape, litchi, and cinnamon), whose structures are shown in Figure 1A. EGCG directly binds to the thick peptidoglycan layer of gram-positive bacteria where it may attack gram-negative bacteria through the production of H_2_O_2_ [21]. EGCG also interferes with bacterial quorum-sensing and affects biofilm formation by reducing autoinducer-2 concentration [22]. In addition, a range of polyphenols inhibits formation of a broad range of amyloids [23,24], not the least FapC from *P. aeruginosa* PAO1 where it commensurately increases biofilm sensitivity to antibiotics [25]. EGCG has been suggested to redirect otherwise amyloidogenic monomers to off-pathway oligomers through interactions between its π-system and the protein backbone [26], though interactions with hydrophobic regions and H-bonding partners also contribute [27,28]. We have recently shown that EGCG inhibits FapC fibrillation even in the presence of bacterial amphiphiles such as rhamnolipids and LPS which otherwise promote fibrillation [29]. PGG is reported to have applications against cancer, diabetes, inflammation, neurodegeneration, viruses, and bacteria [30,31,32,33,34,35,36], but also inhibits aggregation. For example, it binds to the N-terminus and central hydrophobic core of Aβ_1–42_ and Aβ_1–40_, decreasing formation of both oligomers and fibrils [37].

Given the evolutionary optimization of the properties of functional amyloid in contrast to their pathological counterparts, polyphenol inhibitors might impact their fibrillation in a different manner. To investigate this, we have compared the activities of a number of different polyphenols against the fibrillation of FapC wildtype (wt) and truncations thereof. Starting with a broad range of polyphenols, we select EGCG, PGG, baicalein, oleuropein, and procyanidin B2 as the most promising candidates. Based on a broad range of biophysical measurements, we identify EGCG and PGG as the best inhibitors. They redirect FapC to off-pathway oligomers in the form of a compact core surrounded by a flexible shell, remarkably similar to oligomers formed by the pathological amyloid protein α-synuclein. In addition, we use deletion mutants of FapC and peptide arrays to identify which parts of the FapC sequence are involved in these processes. Finally, we show that the inhibitors reduce biofilm formation in *Pseudomonas UK4* overexpressing the *fap*-operon to an extent which correlates with their amyloid-inhibiting properties, highlighting their therapeutic potential against biofilm-mediated infections.

## 2. Materials and Methods

### 2.1. Purification of His-Tagged Recombinant FapC Protein

His-tagged FapC (wt and mutants) were purified as described [17] and stored in 8 M GdmCl. Immediately prior to use, FapC was desalted into 50 mM Tris pH 7.5 with PD-10 columns [17].

### 2.2. Thioflavin T Fibrillation Assay

30 µM FapC was incubated in Tris buffer (50 mM, pH 7.5) with 40 µM ThT and different concentrations of inhibitors (from stocks of 100 mM in DMSO). Note that the low residual amounts of DMSO from the stock solution did not affect fibrillation kinetics (Appendix A). The solutions were monitored in a 96-well plate (Nunc, Thermo Fisher Scientific, Roskilde, Denmark) at 37 °C with 180 rpm orbital shaking together with a sterile 3 mm diameter glass bead, using excitation and emission at 448 and 485 nm, respectively from the plate bottom on a Genios Pro fluorescence plate reader (Tecan, Männedorf, Switzerland). The plate was sealed with clear crystal sealing tape (Hampton Research, Aliso Viejo, CA, USA) to prevent evaporation. Additional experiments at different FapC concentrations (30, 50, and 90.6 µM) and different concentrations of EGCG and Baicalein under quiescent conditions were carried out in a Clariostar plus plate reader (BMG Labtech, Ortenberg, Germany) using the following settings: Excitation: 448.2 nm, emission: 485 nm, gain: 1500, scan mode: Orbital averaging, scan diameter: 4 mm, cycle time: 5 min. Endpoint ThT fluorescence levels were read out directly from the raw data as the average from the 5 h of the fibrillation process, while nucleation (lag) times t_N_ were estimated as the intercept between the base line and the elongation phase.

### 2.3. Reverse Phase High-Performance Liquid Chromatography (RP-HPLC)

100 µM of polyphenol in 50 mM Tris buffer pH 7.5 was incubated on an Eppendorf thermo-shaker TS-100 (BioSan, Riga, Latvia) at 37 °C at 300 rpm. After 0, 1, 2, 4, 8, 16, and 24 h of incubation, 20 µL of the samples were injected onto a C18 column (RP-C18 Luna column, 4.6 mm id × 250 mm, Phenomenex, Cheshire, UK) and elution was monitored by absorption at 270 nm. Materials were eluted using a mobile phase of 30:70 *v*/*v* acetonitrile: Water at a flow rate of 0.5 mL/min for all polyphenols except for the more hydrophobic compound baicalein where the mobile phase consisted of 80:20 *v*/*v* acetonitrile: Water [38].

### 2.4. Circular Dichroism (CD) Spectroscopy

Fibrillated samples were sonicated for 10 min in a Sonorex Digitec ultrasonic bath (Bandelin, Berlin, Germany) and diluted to 15 µM in Tris buffer (50 mM, pH 7.5). Spectra for fibrils and monomers were recorded on a Chirascan CD spectrophotometer (Applied Photophysics, Surrey, UK) in a 1 mm cuvette between 200 and 260 nm with 3-nm bandwidth at 25 °C and three accumulations.

### 2.5. Attenuated Total Reflectance Fourier Transform Infrared Spectroscopy (ATR-FTIR)

To remove unbound polyphenols, fibrillated samples were centrifuged (13,000 rpm for 15 min), supernatants discarded, and the pellet resuspended in the same volume of milliQ water. 2 µL was dried under a stream of nitrogen and spectra recorded on a Tensor 27 FTIR spectrophotometer (Bruker, Billerica, Massachusetts, USA) with a DTGS (deuterated tri-glycine sulfate) Midinfrared detector and Golden Gate single reflection diamond attenuated total reflectance cell (Specac, Kent, UK). Baseline correlation, atmospheric compensation, and second derivative of curves were carried out with OPUS 5.5 software (Bruker, Billerica, Massachusetts, USA).

### 2.6. Seeding Experiments

FapC fibrils and aggregates formed in the presence of polyphenols were pelleted (13,000 rpm for 15 min), supernatant removed, and pellet resuspended in the same volume of milliQ water. The suspension was sonicated with 20% amplitude for 2 min (pulse 5 s on and 5 s off) on a QSonica Sonicator (Q500, Newtown, CT, USA) to obtain short fibrils. 30 µM of FapC monomer was incubated with 2% and 10% (*w*/*w*) of seeds using the Tecan settings above without agitation. Polyphenols were included at 120 µM.

### 2.7. Formic Acid Assay

30 µM FapC monomer was incubated with and without 120 µM PGG as described for the fibrillation assay. After reaching the stationary phase in the fibrillation process, the aggregates were washed twice with milliQ water by centrifugation (13,000 rpm for 15 min) and resuspended in different FA concentrations in water (30%–70%), incubated for 15 min at room temperature, frozen, and freeze dried overnight to remove FA. Dried samples were resuspended in Tris buffer and loaded on 15% acrylamide-bis SDS-PAGE.

### 2.8. Detection of Polyphenols by Nitro Blue Tetrazolium (NBT-Assay)

FapC monomer incubated with EGCG at different time points were loaded on two 12% SDS-PAGE gel with prestained protein ladder (Thermo Scientific, Roskilde, Denmark). One gel was stained with Coomassie Brilliant Blue to visualize protein bands. The other gel was electroblotted onto PVDF membrane (Merck Millipore, Darmstadt, Germany) for 1 h at 200 mA, after which the membrane was stained with NBT solution (0.6 mg/mL NBT, 2 M potassium glycinate at pH 10) in the dark for 45 min. The membrane was washed with 0.64 M boric acid at pH 9.0 and then with milliQ water and scanned with an image scanner.

### 2.9. Celluspots™ Peptide Arrays

Peptide arrays were prepared as described [39]. Each peptide array was blocked with 25 mL blocking solution (PBS with 0.1% Tween-20 (PBS-T) containing 3% (*w*/*v*) whey protein powder) in a Nunc™ tube on a rolling board overnight at 4 °C. Two arrays were incubated with or without 0.05 mg/mL EGCG (dissolved in PBS-T pH 7.5) for 6 h on a rolling board at room temperature (25 °C), washed three times with PBS-T and three times in milliQ water. Subsequently the plates were submerged in NBT-T (0.6 mg/mL NBT, 2 M potassium glycinate, 0.1% Tween-20, pH 10) solution for 45 min in the dark, washed with boric acid solution and milliQ water, left to dry for ~5 min and imaged using a scanner. Results were converted to signal intensities with the protein array analyzer toolbox in Image J.

### 2.10. Size Exclusion Chromatography (SEC)

Samples were centrifuged (13,000 rpm for 10 min) and the supernatants were loaded on a 24 mL Superose 6 10/300 column (GE Healthcare, Brondby, Denmark) at a flow rate of 0.5 mL/min and absorbance were recorded at 280 nm (Agilent 1260 Analytical UV cell). Oligomers were purified by fractionation of the output current and concentrated using an Amicon Ultra-15 centrifugal filter with 15 kDa molecular weight cut-off.

### 2.11. Small-Angle X-ray Scattering (SAXS)

Oligomer samples were purified by SEC and measured together with freshly desalted samples of FapC, FapC with TCEP, the mutant ΔR1R2R3 with and without TCEP. Buffers were measured for background subtraction. The protein concentration determined at 280 nm with a NanoDrop™ 1000 spectrophotometer (Thermo Scientific, Roskilde, Denmark) based on the extinction coefficient of FapC monomer (10,095 M^−1^ cm^−1^). All experiments were carried out on an in-house modified NanoSTAR SAXS instrument (Bruker AXS) [40] with a Ga liquid metal yet source (Excillum) [41] and scatterless slits [42]. The SUPERSAXS package (C.L.P. Oliveira and J.S.P., unpublished) was used for initial data treatment, buffer subtraction, and absolute scale normalization using water. All data are displayed as a function of the modulus of the scattering vector *q* = 4π/λ sin(θ), with 2θ being the scattering angle and λ = 1.34 Å. The home-written program WIFT [43] was used to carry out indirect Fourier transformation (IFT) [44] and obtain the pair distance distribution function (*p*(*r*)), which is a histogram of distances in the sample weighted by the excess electron density. The IFT gives information on the average size of molecules in the solution in terms of radius of gyration (*R*_g_) and also on the molar mass (*Mw*) through the forward scattering (*I*(*q* = 0)), which can be converted into *Mw* using a protein contrast value of ∆ρ_m_ = 2.00 × 10^10^ cm/g and the relation M = *I*(0)/(*c* ∆ρ_m_^2^), where c is the protein concentration. The mass can be converted into an aggregation number of the protein (*N_agg_*) by dividing by the mass of the monomeric protein. To model the oligomers, we used a core-shell model with a compact core surrounded by a shell of flexible protein as described [45]. Modelling was done on an absolute scale to obtain *N_agg_*. Smearing of the electron density was used in both the core (*σ*_in_) and the shell (*σ*_out_) using a smearing function of $e^{-q^{2}\sigma /2}$. Furthermore, the core was described as a prolate ellipsoid with the long axis being multiplied by *ε* and the core radius was set to *R_cor_*_e_ = 2 *σ*_in_ and the shell radius to *R_shell_* = 2 *σ*_in_ + 2 *σ*_out_. The total size of the complex was determined by the diameter = 8 *σ*_in_ + 4 *σ*_out_ along the short axis of the ellipsoidal structure. The fraction of mass in the shell (*f*_shell_) was determined from the size of the core compared to the shell and used to calculate the polymer scattering arising from the flexible region in the shell. Scattering contributions from added polyphenols were ignored due to their insignificant contributions.

### 2.12. Measuring Amount osf Biofilm with Crystal Violet

A single colony of *Pseudomonas sp. UK4* (wt, ∆fap, and pfap) was transferred to the LB medium at 28 °C, 180 rpm for overnight growth. The solutions were then diluted to OD_600_~0.5 in the presence of ampicillin (100 µM) to avoid contamination, and 160 µL hereof added to each well of a 96-well plate. Peg lids (Nunc-Tsp, Thermo Scientific, Roskilde, Denmark) were inserted in the plate wells for 1 h to initiate biofilm growth on the peg lids, after which the peg lids were transferred to another fresh LB medium (160 µL per well) with 100 µM polyphenol for 48 h incubation, resulting in growth of biofilm on pegs. The peg lids were then washed in milliQ water to remove planktonic bacteria and dried for 1 h at room temperature, after which they were submerged in a new plate with 160 µL per well Gram’s crystal violet solution for 15 min. The peg lid was washed twice with milliQ water to remove excess stain and bound crystal violet was released by incubation of the peg lids in 33% *v*/*v* acetic acid (glacial acetic acid diluted with milliQ water) for 30 min at room temperature. The absorbance intensity at 590 nm was used as a measure of the amount of biofilm on the pegs.

### 2.13. Confocal Microscopy

One colony of *Pseudomonas sp. UK4* (wt, ∆fap, and pfap) was transferred to the LB medium for overnight growth at 28 °C, 180 rpm, diluted to OD_600_~0.5, and transferred to an IBIDI^®^ μ-slide VI^0.4^ uncoated channel slide. After 1 h incubation, the bacteria were gently washed with fresh LB medium, after which the LB medium with 100 µM of PGG and EGCG was circulated through the chambers at a flow rate of 2 mL/h for 48 h (Harvard PHD 2000 Infuse/Withdraw Syringe Pump) to enable formation of biofilm under dynamic conditions. After washing the chambers with PBS and staining with live/dead kit using Cyto9 and propidium iodide, bacteria were imaged with LSM 510 scanning confocal microscope (Zeiss GmbH, Jena, Germany).

## 3. Results

### 3.1. Polyphenols Inhibit FapC Fibrillation by Forming Non-Amyloid Aggregates

As an initial step, we compare the ability of ten different polyphenols to inhibit FapC fibrillation. These ten compounds are PGG, EGCG, oleuropein, baicalein, procyanidin-B2, verbascoside, 3-hydroxytyrosol, epicatechin, catechin, and gallic acid (Appendix A). All ten compounds contain multiple vicinal hydroxyl groups, ranging from single aromatic rings (gallic acid and 3-hydroxy-terosol) to complex systems linked through a glucose ring (PGG); furthermore, gallic acid and catechin/epicatechin are subscaffolds of EGCG, procyanidin, and PGG. We incubated 30 µM wt FapC monomer with 10–200 µM polyphenols (Appendix A), monitoring fibril formation with the amyloid-binding dye Thioflavin T (ThT, representative runs in Figure 1B,C). All compounds decrease ThT end-point emission in a dose-dependent fashion (Figure 1D, Appendix A), with PGG showing the strongest effect, followed by baicalein, EGCG, procyanidin B2, and oleuropein. The remaining five compounds showed significantly weaker effects, indicating that a reduction in compound size below a certain level of complexity leads to a loss of activity. Accordingly, we proceed with the five best polyphenols.

We used RP-HPLC to assess compound stability over 24 h at 37 °C pH 7 (Appendix A). While the population of the starting compound declined by 40%–70% over 24 h, the compounds were largely stable (2%–25% loss) over the first 2 h, corresponding to the lag phase of FapC fibrillation (Figure 1). Thus, the compounds are essentially intact during this critical time period where fibrillation would otherwise start to occur. Furthermore, the compounds form stable adducts with FapC within the first hour of incubation (see below).

SDS-PAGE analysis of the soluble fraction from the ThT end-point samples revealed high molecular weight bands only for PGG and EGCG, but in all cases only a small amount of monomeric FapC (226 residues for the mature protein, molecular weight 22.5 kDa but migrating with an apparent weight of ≈31 kDa) (Figure 1E). Larger aggregates (fibrils or other kinds of aggregate probably produced by effect of polyphenols) will not enter the gel. Thus, most of FapC must assemble into aggregates of some kind even in the presence of polyphenols. Note that a ~25 kDa band was visible migrating below the monomer. Previous work studying proteins recombinantly expressed in *E. coli* and purified by Ni-NTA chromatography attribute this 25 kDa band to the protein SlyD, whose high content of His residues in its C-terminus leads to a relatively high affinity for the Ni-NTA column used during purification of both FapC and CsgA [46].

Reduction of ThT fluorescence could be caused either by a genuine reduction of the amount of fibrils formed or by the displacement of ThT from amyloid by polyphenols, i.e., a misleading artifact. To resolve this, we analyzed the samples at the end of the fibrillation process using different structural techniques. According to the far-UV circular dichroism (CD), all samples had significant β-sheet structures with a minimum around 218 nm, though with reduced intensity, particularly for PGG and EGCG (Figure 1F). For complementary data on the secondary structure, we used ATR-FTIR (attenuated total reflectance Fourier transform infrared spectroscopy). The 2nd derivative spectrum (Figure 1G) in the control (compound-free) FapC fibrils highlighted the characteristic amyloid peak around 1623 cm^−1^ as well as a smaller peak at 1663 cm^−1^ indicative of antiparallel β sheets [47]. These peaks almost disappear for PGG and EGCG and diminish to some extent for oleuropein, procyanidin B2, baicalein, and control. CD and FTIR thus confirm reduction in β-sheet content by both PGG and EGCG; more specifically, we observe a strong linear correlation between end-point ThT intensities and CD as well as FTIR signals (Figure 1H). Therefore, the endpoint ThT fluorescence is an accurate reporter of the amyloid amounts and can be used to calculate the apparent IC_50_ value where the end-point ThT level has been reduced to 50% of the original value (Figure 1D). This ranked the compounds as PGG > baicalein ≈ EGCG > procyanidin B2 ≈ oleuropein in terms of inhibitory potency.

Consistent with the observation of high-molecular weight bands in Figure 1E, TEM of samples with PGG and EGCG shows small spherical species with diameters around 19 ± 1 nm whereas the other three polyphenols give rise to fibrils with an average width of 13 ± 1 nm width, surrounded by more amorphous aggregates (Figure 2).

The formation of non-amyloid aggregates with PGG and EGCG, whose presence is not captured by our ThT assay, significantly complicates a kinetic analysis of the data. Although the effect of inhibition can be attributed purely to a depletion of the monomer concentration into non-amyloid aggregates over the course of the aggregation reaction, the concentration of non-amyloid aggregates over time, and thus also the monomer concentration, is not known. EGCG shows a longer lag phase than baicalein (Figure 1C) but reaches the same endpoint ThT level. This may reflect the different kinetics of the formation of the non-amyloid species. By examining the early time behavior of the data (Figure 1C) it is clear that amyloid forms at comparable rates to the inhibitor-free case in the presence of oleuropein and procyanidin, but forms significantly more slowly in the presence of baicalein, PGG and EGCG. Assuming that the inhibitory effect is purely an indirect one via depletion of monomers, this finding indicates that a significant level of non-amyloid structures has been reached after 1 h in the presence of PGG, EGCG, and baicalein.

### 3.2. FapC Aggregates Formed with Polyphenols Show High Stability but Reduced Seeding Capacity

Given that PGG was the best inhibitor of FapC amyloid formation, we decided to investigate the stability of the non-fibrillar aggregates formed in the presence of this polyphenol. Many pathological fibrils dissolve at very low concentrations of the potent denaturant formic acid (FA); in contrast, functional amyloid typically requires > 80% FA for complete dissolution [11]. The aggregates were spun down and exposed to different concentrations of FA, lyophilized (to remove FA), and analyzed by SDS-PAGE. There is a marked difference between aggregates in samples with PGG and fibrils in polyphenol-free samples (Appendix A). With PGG, aggregates resist up to 40% FA, above which they gradually dissociate to a mixture of monomers, dimers, and higher-order oligomeric species. In contrast, FapC fibrils only form monomers and a very small amount of dimers. The monomers already start to form from 30% FA onwards, emphasizing the difference between the two aggregate types and indicating a strong stabilization of non-amyloid aggregates by PGG.

To investigate whether the polyphenols inhibit growth of FapC fibrils, 10% (*v*/*v*) sonicated FapC fibrils were used to seed solutions of monomeric FapC with and without polyphenols (raw data in Appendix A, summarized in Figure 3A). At these high seed concentrations, the kinetics are dominated almost purely by growth of existing fibrils and nucleation of free monomers is essentially negligible. Both alone and in the presence of the different polyphenols, except for PGG, the seeds completely abolish the lag phase of normal FapC fibrillation. PGG completely suppress the ThT signal growth. The endpoint fluorescence of these samples was consistently lower than in the absence of seeds, indicating that the non-amyloid species are formed rapidly compared to the timescale of fibril elongation, even in the absence of nucleation.

The ensuing aggregates were then used in a second assay to evaluate the seeding capacity of aggregates formed in the presence of inhibitors. Each sample was spun down after fibrillation to remove unbound polyphenols and monomeric FapC inhibitors, after which the aggregates were resuspended in water, sonicated, and added to 2% and 10% (*v*/*v*) of FapC monomer in the absence of additional polyphenols (Figure 3B,C). In all cases, the seeds eliminate the lag phase but the rise in ThT fluorescence is reduced to a small extent in a ranking order consistent with the previous inhibitor performance of the polyphenols. Thus, the end-level aggregates formed by the different inhibitors do retain the ability to promote FapC fibrillation but to a slightly reduced extent.

### 3.3. Increased Nucleation Times Increase the Inhibitory Effects of Polyphenols

For comparison we turned to the closely related FapC-PAO1 from *P. aeruginosa*, which differs from FapC-UK4 in having a significantly longer linker region between repeats 2 and 3. Polyphenol effects were broadly similar to those on FapC-UK4. FapC-PAO1 has a 3–4 fold longer lag phase than the FapC UK4 without inhibitor (Figure 4A) and this is further increased by baicalein, PGG, and EGCG. It is possible that the increased lag phase of FapC-PAO1 may enhance its susceptibility to polyphenols, just as the most slowly aggregating FapC-UK4 mutant, but not the faster variants, inhibits aggregation of α-synuclein [48]. However, the end-point ThT fluorescence and nucleation (lag) times t_N_ (normalized to values in the absence of polyphenols) are broadly similar between the two FapC variants and once again show that PGG, EGCG, and baicalein are the most efficient inhibitors. Further, PGG, EGCG, and baicalein keep FapC monomeric or lead to off-pathway oligomers (Figure 4B).

To expand on these initial observations, we turned to mutants of FapC-UK4 in which one, two, or three of the imperfect repeats had been removed. The deletion mutants allow us to investigate if some regions of FapC are particularly important for interactions with the polyphenols (ThT time profiles in Appendix A). The lag phase increases particularly in mutants lacking the R3 repeat (Figure 5A). This indicates that R3 is the most aggregation prone repeat, consistent with previous results [17]. PGG and EGCG considerably reduced the intensity of ThT-emission and increased t_N_ for all mutants (Figure 5B,C).

We used SDS-PAGE to monitor the amount of soluble species of the different FapC-UK4 mutants left at the end of aggregation in the presence of polyphenols (Appendix A, summarized in Figure 5D,E). This provided further evidence for the link between longer nucleation times and greater effect of inhibitors. Overall, the longer the lag phase, the greater the tendency for FapC-UK4 to be maintained as monomer or off-pathway oligomers, particularly in the presence of PGG and EGCG. This is clearly seen for mutants lacking repeat R3, most strongly for the mutant lacking all three repeats which failed to fibrillate both according to ThT (Appendix A) and TEM (Appendix A). TEM images of mutants incubated without and with EGCG (Appendix A) confirmed that EGCG drove formation of large amounts of oligomers in R3-free mutants according to SDS-PAGE. To increase the lag phase further, we incubated wt FapC-UK4 without agitation (Appendix A) and analyzed the content of the end-phase samples by SDS-PAGE (Appendix A). Quiescence increased the lag phase of the control sample from 1 to 5 h. Comparison with Figure 1B and 1D shows that quiescence also leads to a marked decrease in normalized ThT fluorescence for all compounds (Appendix A) as well as a significant increase in the concentration of both oligomer and (unpolymerized) monomer, particularly for EGCG (Appendix A).

Note that deletion of R2 in the ∆R2,3 mutant prolongs the lag phase more than the deletion of R1 in ∆R1,3 FapC (Figure 5A). The SDS-PAGE of FapC variants incubated with EGCG also show stronger oligomer and monomer bands for ∆R2,3 than ∆R1,3 (Figure 5D) that could be due to longer lag phase of ∆R2,3. The considerable amount of oligomer formation of mutant ∆R1,2,3 in the presence of PGG and EGCG (Appendix A—∆R1,2,3 and Appendix A) indicates that the linker regions provide interaction sites for polyphenols (see below).

### 3.4. PGG and EGCG Bind to FapC Monomers and Early Oligomers to Inhibit Fibril Formation

Our inhibition results prompted a closer investigation of their interaction with monomers and soluble aggregates of FapC. We exploited that the compound nitroblue tetrazolium (NBT) is converted to dark-purple formazan by compounds containing quinones such as EGCG [25]. Accordingly, we concentrate on EGCG in the following. Oligomers were purified by SEC, separated on SDS-PAGE, transferred to PVDF membranes, and finally stained with NBT (Figure 6A). There was a good correspondence between species seen by Coomassie staining and NBT, indicating that all species were able to bind to EGCG. However, the smaller species could conceivably derive simply from dissociation of the oligomers. To address whether monomers also react with EGCG, we followed the evolution of different species over time with NBT staining, which highlights both monomeric and oligomeric species formed over the first 8 h (Figure 6B). Nevertheless, the oligomers show up more clearly by NBT than Coomassie staining, indicating stronger interactions with these species. There is a gradual fading of the NBT color over time which may also be due to slow oxidative modifications of EGCG.

Finally, we probed the binding of EGCG to different parts of the 226-residue mature FapC-UK4 sequence (i.e., without the 24-residue signal peptide sequence). For this we used a peptide microarray that displays the whole protein as 10-residue peptides displaced five residues along the sequence, giving each peptide spot a five-residue overlap with the next spot. The extent of EGCG binding to each spot is quantified by NBT-staining followed by densitometric analysis with Image J. A control array that was not exposed to EGCG did not develop any color with NBT (Appendix A). The intensity of EGCG binding varied significantly across the probed peptides (Figure 6C), showing particular affinity for regions corresponding to the very N-terminal part, a highly amyloidogenic sequence in repeats R2 and R3 (but not R1) based on Rosetta energy calculations [39], linkers between R1–R2 and R2–R3, and the very C-terminal part. There was no clear correlation with hydrophobicity or aromaticity (data not shown). The strong binding to linker regions rationalizes the effect of EGCG on the triple mutant of FapC (∆R1,2,3) which only includes the linker regions; however it does not explain the variation in oligomer production by different FapC variants (Figure 5D), as they all contain linker regions. Presumably it is the combination of EGCG binding sites and their accessibility through prolonged lag phases that facilitates oligomer accumulation.

### 3.5. Polyphenols Have Different Potential for Assembly of Off-Pathway Oligomers

As shown by SDS-PAGE (Figure 4B), the polyphenols lead to accumulation of oligomers to different extents, but it was not *a priori* clear whether the size and shape of these oligomers differed among the different polyphenols. Accordingly, we quantified oligomer concentration by SEC after 1 h of incubation with polyphenols. There was a direct correlation between the inhibitory effect and oligomer amount (Figure 7A). PGG and EGCG led to significantly larger oligomer peaks (eluting around 25 mL) than the other polyphenols which in turn outperformed the control. To monitor the oligomer development over time, we analyzed population sizes with SEC after different incubation time points. We compare the best inhibitor (PGG), the worst (oleuropein), and control (Figure 7B–D). In all cases, the oligomer concentration declines over time and disappears completely after 24 h. PGG leads to the largest oligomeric species which declines slowly in intensity over 24 h, while oleuropein and control only lead to small amounts of a species which is probably a small oligomeric (e.g., dimer and trimer) mixture, eluting as a shoulder on the monomer peak. This peak disappears within a few hours, most likely it is being incorporated into growing fibrils. The peaks eluting after monomer in SEC are attributed to polyphenols due to their long retention and lack of signal on SDS-PAGE (data not shown). The high signal intensity of the PGG oligomers (significantly higher than that of the monomers) can be ascribed to the PGG bound to the oligomers, cf. NBT-staining (Figure 5A).

To investigate the oligomer structure, we purified oligomers prepared in the presence of PGG, EGCG, and Baicalein and analyzed them by TEM, SAXS, and CD. TEM images displayed remarkably monodisperse spherical oligomers with a diameter around 18 nm in the presence of inhibitors while the size of the control oligomers was around 10 nm (Figure 8A control). The oligomers show the same random coil spectrum as monomeric FapC (Figure 8B). All three off-pathway oligomers showed a highly similar SAXS scattering pattern (Figure 8C,D). The pair-distance distribution function (*p*(r)), which is a histogram over distances between pair of points within the particle, was determined by indirect Fourier transformation (IFT). The main feature between 0 and 20 nm indicated that the PGG, EGCG, and baicalein oligomers all had a somewhat compact structure and a radius of gyration (*R_g_*) for the whole oligomer (core and shell) of 10.1 ± 0.2, 7.2 ± 0.1, and 10.0 ± 0.2 nm, respectively (Table 1). We were able to fit to the data with a model developed for α-synuclein oligomers [45], in which a compact prolate ellipsoidal core is surrounded by a shell of flexible protein (Figure 8E). This model could describe all three oligomers and gave short axis diameters of 27.2 ± 0.5, 25.6 ± 0.3, and 25.8 ± 0.4 nm and similar aggregations numbers (*N_agg_*) of 7.6 ± 0.2, 6.7 ± 0.1, and 6.7 ± 0.1 for PGG, EGCG, and baicalein, respectively. No change in the structure of the oligomers was seen when they were incubated at 37 °C for 16 h (data not shown). The thickness of the flexible shell was similar for all three oligomers with 0.31–0.36 of the total protein mass being located in this region. The shown values of *N*_agg_ in Table 1 were determined from the determined *p*(*r*) function, and they are similar to values obtained by the modeling (data not shown).

The ∆R1R2R3 mutant lacking all three imperfect repeats also formed oligomers in the presence of PGG, consistent with its behavior with EGCG (Figure 5D). SAXS analysis of this oligomer revealed a similar core-shell structure as the other oligomers. Like the other oligomers, ~ one third of the protein was found in the shell with flexible protein (*f*_shell_ = 0.27), but the oligomer was considerably smaller than that formed by full-length FapC, consisting of less than five monomers and a short axis diameter of 19.5 ± 0.7 nm (Table 1).

For comparison, we also analyzed the freshly desalted samples of FapC, and the mutant ∆R1R2R3 both with and without TCEP. The SAXS data (Appendix A) are qualitatively different from those of the oligomers, with a much less pronounced decay for increasing *q*. In the presence of TCEP, the IFT gave aggregation numbers of 1.4 and 1.8 for FapC and ∆R1R2R3, respectively, whereas they were higher without TCEP (4.4 and 2.7, respectively). The radii of gyration were 5.9 ± 0.1 nm for both proteins with TCEP, and 10.1 ± 0.2 and 7.0 ± 0.1 nm without TCEP for FapC and ∆R1R2R3, respectively. The data could be fitted well with the random coil model, which shows that the aggregates in the freshly desalted samples are different from those of the purified oligomers, and do not have a compact core. The model fitting gave similar results for aggregation numbers and radii of gyration.

### 3.6. The Polyphenols Reduce the Extent of Pseudomonas-UK4 Biofilm Formation

Finally, we analyzed the effect of inhibitors on biofilm formation of *Pseudomonas fluorescens UK4* which produces FapC-UK4. We have previously shown that the biofilm of this strain is strengthened mechanically by the production of amyloid [12]; unlike *P. aeruginosa*, amyloid formation is not accompanied by alginate formation which can otherwise complicate biofilm quantification. We incubated the *fap* overexpressing strain (pFap) in the presence of polyphenols, using either 96-well peg-lid plates to quantify the extent of biofilm formation (staining by crystal violet) (Figure 9A) or flow chambers for confocal microscopy images. We see a remarkably clear correlation between the biofilm mass and ThT maximum emission (Figure 9B); the lower the emission (and thus the greater the inhibition of aggregation), the lower the amount of biofilm formed. Confocal images (Figure 9C) in the absence of polyphenols show that the pFap strain made more biofilm than wt and the ∆Fap strain lacking the *fap* operon, where the bacteria show much less clumping. This emphasizes the significant role of FapC amyloid in biofilm formation [16]. However, in the presence of PGG and EGCG, the bacteria are more dispersed and there is a greater proportion of dead cells (which stain red in Figure 9A).

## 4. Discussion

A major challenge in the treatment of bacterial infections is the formation of a self-produced biofilm matrix which protects the resident bacteria against conventional antibiotic, mechanical, and chemical stresses. Several strategies have been used to develop therapeutic compounds to combat these biofilms. Amyloid is known to strengthen this biofilm, as illustrated by the role of FapC in promoting biofilm in *Pseudomonas* [16]. By inhibiting their formation, compounds such as polyphenols (many of which are available in common dietary components) provide a potential tool in this combat. Hence, we have carried out a study of the effect of five selected plant polyphenols on FapC fibrillation. Prior to that, we established that smaller compounds such as gallic acid, catechin, epicatechin, and 3-hydroxytyrosol, which represent subscaffolds of the larger polyphenols, performed poorly compared to the more complex polyphenols. This indicates cooperativity effects from combining several functional groups close in space.

A dose-response ThT-emission profile identified PGG (IC_50_~26 µM) as the best inhibitor, closely followed by EGCG. Moreover, an increase in the nucleation time t_N_ for both PGG and EGCG were observed, also consistent with a depletion of monomer to a non-ThT-active phase. The aggregates that accumulated in this process consisted of oligomers and larger non-fibrillar aggregates which were not seen in the absence of these polyphenols. The oligomers were remarkably stable towards formic acid, but did not seed fibrillation, consistent with their random coil structure and tendency to bind polyphenols. Aggregates formed in the presence of inhibitors showed a slightly reduced ability to seed compared to *bona fide* FapC fibrils, in particular aggregates with PGG, which also showed low β-sheet content. The polyphenols’ effects are not limited to the prevention of fibrillation: We have previously shown that EGCG can remodel FapC fibrils into non-amyloid aggregates [25].

### 4.1. EGCG Targets Hot Spots in FapC, Leading to Off-Pathway Unstructured Oligomers

Different truncation mutants of FapC lacking one or more of its imperfect were incubated with polyphenols to address which parts are important in inhibition. A cocktail of different assays confirmed R3 as the most important repeat whose deletion significantly extended the fibrillation lag phase, followed by R2. PGG and EGCG consistently ranked as the two polyphenols with the greatest effect on the mutants’ properties. The mutant ∆R1,2,3 lacking all three repeats underwent complete conversion to oligomers, indicating that the linker region is also able to interact with EGCG. This is gratifyingly consistent with our peptide array data which identifies “hot spots” for interaction with EGCG in the linker region as well as repeats R2 and R3 which (unlike R1) contain the “GVNVAA” sequence, previously identified as highly amyloidogenic in FapC-PAO1 [39]. The high hydrophobicity score of this hexapeptide could provide a high potential for interaction of polyphenols in the EGCG. Thus, the most amyloidogenic part and the linker region of monomer can interact with EGCG and significantly inhibit the fibrillation process. Binding of a relatively large aromatic molecule such as PGG to FapC monomers or early oligomers, confirmed by SEC and SDS-PAGE/NBT, may also sterically block attachment of additional FapC monomers. The oligomers are formed within the first h of incubation, preventing nucleation by the formation of a compact core surrounded by a flexible shell that consists of ~ one third of the entire protein. The CD of separated off-pathway oligomers displayed random coil structures such as the monomer, emphasizing that these species are not β-sheet and definitely non-fibrillar. The polyphenol-stabilized oligomers are larger than the control oligomers formed in the absence of polyphenols and are estimated to consist of ~7 monomers with short axis diameter around 25–27 nm, significantly larger than the small oligomeric species formed in the absence of polyphenols. In contrast, the polyphenol-free oligomers have a diameter estimated by TEM to be 10 ± 0.2 nm (cf. Figure 8A), which indicates that they contain a significantly lower number of monomers, probably around 2–3.

### 4.2. Biofilm may be Targeted through Their Amyloid Component

We also demonstrated a strong correlation between amyloid inhibition and the ability to reduce biofilm formation in *Pseudomonas sp.UK4*, using a FapC overexpression strain, which showed high levels of amyloid production and consequent biofilm formation in the absence of these polyphenols. This biofilm reduction is likely to have direct consequences for the penetrance of antibiotics. Similarly, treatment of *P. aeroginosa* biofilm by EGCG increased its susceptibility to antibiotics and decreased its stiffness [25] (though high levels of alginate production complicated analysis of biofilm production in this species), while the polyphenol tannic acid inhibits the ability of *S. aureus* to colonize surfaces [49]. While our polyphenols effectively inhibit biofilm formation in vivo, their chemical stability remains an issue, given that the concentration of the original compound declines significantly over a 24-h period. However, interactions with existing amyloid components (particularly chemical modifications) occur within the first hour (as detected by our NBT assay). Given that fibrillation of our functional proteins shows a lag time of several hours, there is clearly ample time for the polyphenols to interact with their protein targets before fibrillation takes off in earnest. In practice, regular supplies of fresh polyphenols (e.g., on a daily basis) would likely be sufficient to ensure full efficacy against unwanted bacterial protein fibrillation. Furthermore, it should be noted that their chemical reactivity is connected to their anti-oxidant potential, most likely through resonance stabilization of oxidation-induced free radical states [50,51]. PGG scavenges reactive oxygen species such as superoxide (O_2_^−^) and H_2_O_2_, enhancing resistance of neuronal cells to hydrogen peroxide [52,53]. Use of polyphenols against biofilm is also encouraged by the observations that e.g., EGCG [54] and oleuropein [55] do not show toxicity issues in animal models and EGCG has even progressed to clinical trials for use against Alzheimer’s Disease [23].

In conclusion, our investigations suggest that phenol-rich compounds found plentifully in fruits and plants have a substantial inhibitory effect on fibrillation of FapC, the major component of amyloid structure in the biofilm of many different *Pseudomonas* strains. Further investigation of this inhibitory effect revealed that strong interactions with the linker region and the most amyloidogenic parts of the monomer led to off-pathway oligomers which do not propagate fibrils. We speculate that R2, R3, and the C-terminal part of FapC may form the dense core of oligomer while the flexible shell consists of R1 and the N-terminal region, though this will have to be substantiated with e.g., hydrogen-deuterium exchange mass spectrometry [56]. These data highlight the role of polyphenols, especially PGG, as promising compounds for controlling biofilm-related infections. These findings may have repercussions for our understanding of how polyphenols affect fibrillation of amyloidogenic proteins in neurodegenerative diseases.

## Figures and Tables

**Figure 1 biomolecules-09-00659-f001:**
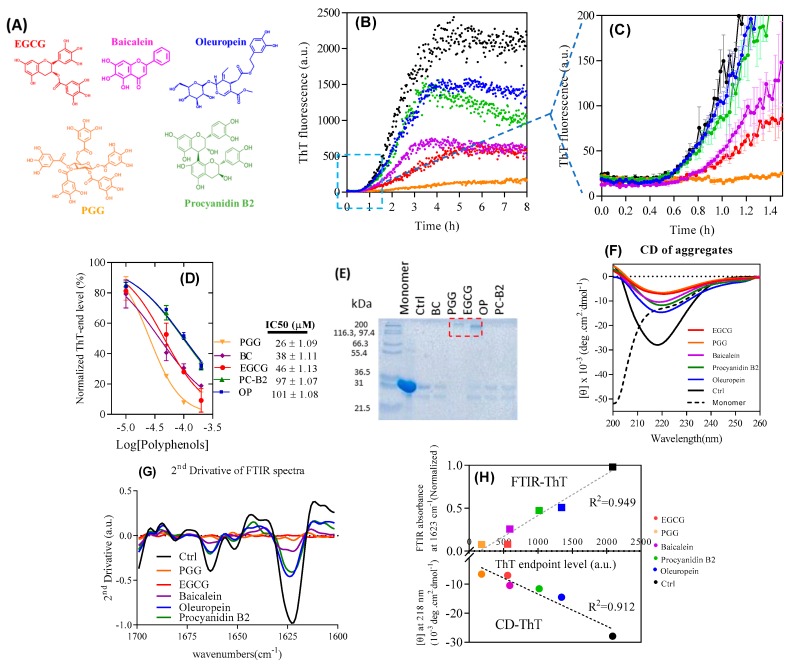
Effect of polyphenols on fibrillation-prone protein (FapC)-*UK4* fibrillation. (**A**) Structures of compounds used in this study. Compound colors correspond to the label colors in subsequent graphs; (**B**) time course of fibrillation of 30 µM FapC with 100 µM polyphenols at pH 7.5 and 37 °C monitored by Thioflavin T (ThT) fluorescence (**C**) and zoom-in in (A); (**D**) IC50 values for inhibition of fibrillation based on data in panel b; (**E**) SDS-PAGE of samples after incubation with polyphenols. Red box highlights the oligomer band; (**F**) Circular Dichroism (CD) and (**G**) ATR-FTIR of samples from panel A after the end of incubation. 1623 and 1663 cm^−1^ peaks are attributed to β-sheet; (**H**) correlation between ATR-FTIR/CD and ThT endpoint levels.

**Figure 2 biomolecules-09-00659-f002:**
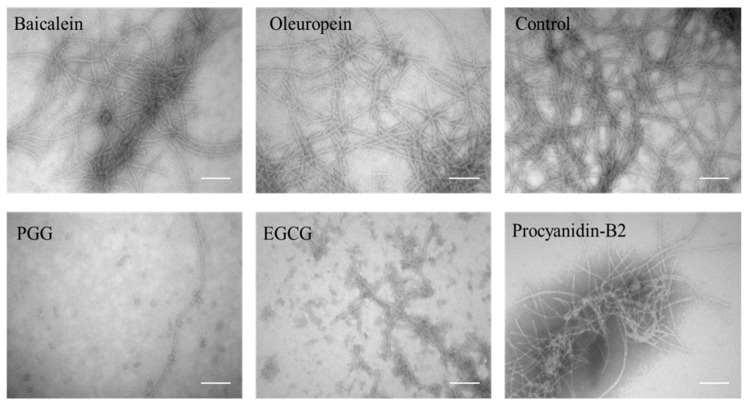
TEM image of aggregates after incubation with polyphenols. Scale bars are 200 nm.

**Figure 3 biomolecules-09-00659-f003:**
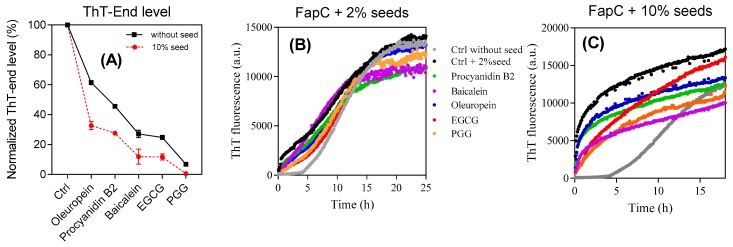
Seeding experiments to test the ability of different aggregates to promote fibrillation of monomeric FapC-*UK4*. (**A**) Seeds: FapC fibrils. Solution: Monomeric FapC (30 µM) together with polyphenols (120 µM); (**B**,**C**) seeds: Aggregates of FapC formed in the presence of 2% and 10% (*v*/*v*) polyphenols and spun down. Solution: Monomeric FapC (30 µM). The indicated labelling scheme applies to both (**B**,**C**).

**Figure 4 biomolecules-09-00659-f004:**
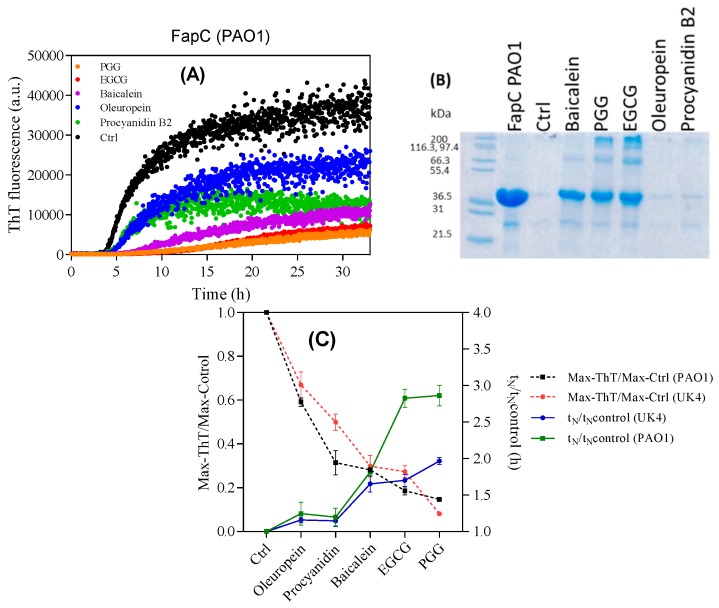
Incubation of FapC from *P. aeruginosa* with different polyphenols. (**A**) ThT time profile of 30 µM FapC-PAO1 incubated with 100 µM polyphenols with shaking at 37 °C; (**B**) SDS-PAGE analysis of end-point samples from panel A. Epigallocatechin gallate (EGCG), penta-*O*-galloyl-β-d-glucose (PGG), and Baicalein maintain FapC in the monomeric or oligomeric state and inhibit fibrillation; (**C**) normalized end-point ThT levels and lag times of FapC fibrillation with different polyphenols.

**Figure 5 biomolecules-09-00659-f005:**
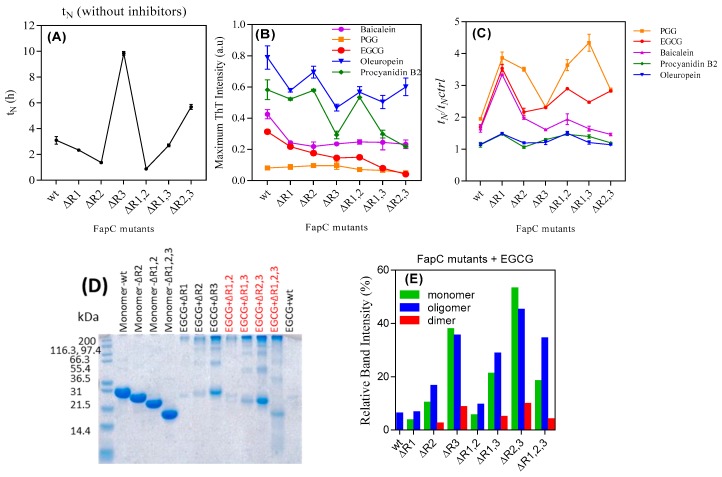
The inhibitory effect of polyphenols on the fibrillation of different FapC mutants. 30 µM of purified mutants of FapC-UK4 was incubated with 100 µM polyphenols in Tris buffer (pH 7.5) with shaking at 37 °C. (**A**) Absolute lag time (t_N_) in the absence of polyphenols, (**B**) normalized maximum ThT emission of different mutants in the presence of inhibitors, (**C**) relative half time (t_N_/t_N,control_); (**D**) SDS-PAGE of different mutants incubated with EGCG; (**E**) quantification of monomer, dimer, and oligomer bands from panel e using Image J.

**Figure 6 biomolecules-09-00659-f006:**
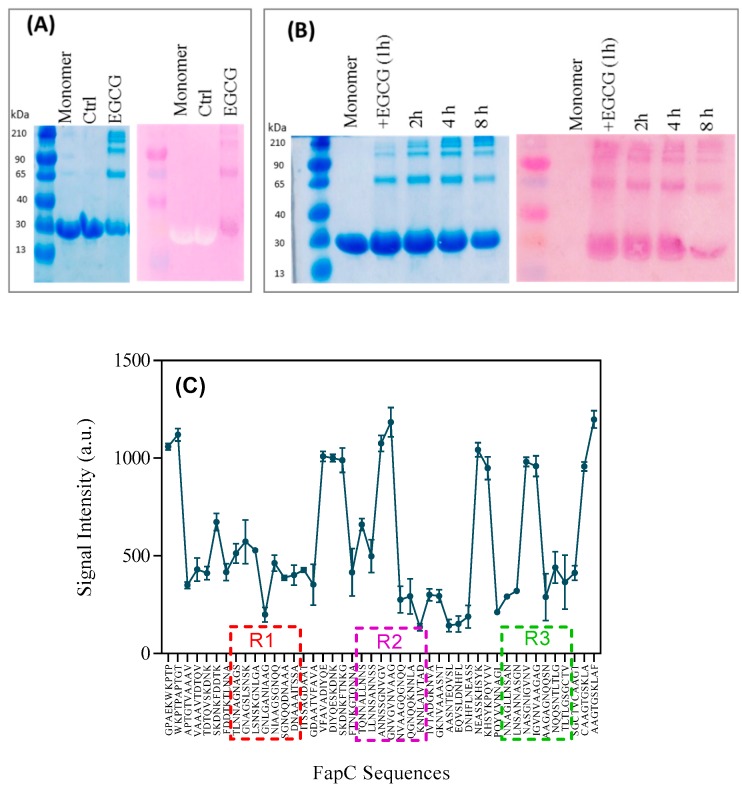
NBT-assay demonstrating polyphenol interactions with FapC. (**A**) Right: SDS-PAGE of oligomers separated by SEC, left: A copy of the same SDS-PAGE gel was electroblotted onto PVDF membrane and stained with NBT; (**B**) time study of the EGCG-FapC interaction using (right) SDS-PAGE) and (left) NBT-stained PVDF membrane, respectively. Samples were taken out at the indicated times and frozen before running on the gel; (**C**) signal intensities of peptide array displaying peptides corresponding to different sequences of FapC, incubated with EGCG, and stained with NBT to identify interacting sequences.

**Figure 7 biomolecules-09-00659-f007:**
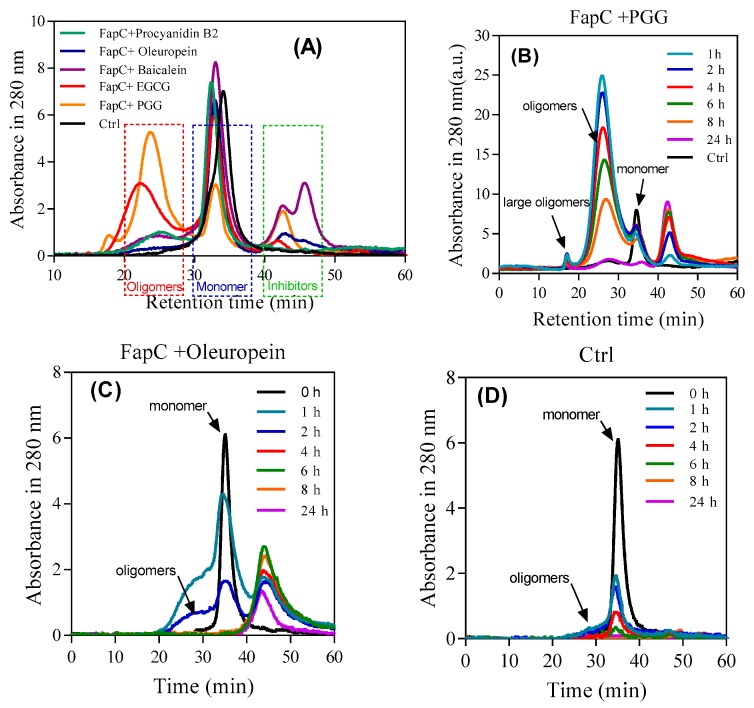
SEC- and SDS-PAGE-based analysis of FapC oligomers formed in the presence of polyphenols. (**A**) After 1 h incubation, samples were separated by SEC and the indicated oligomer fractions pooled and run on SDS-PAGE; (**B**–**D**) time course of oligomer formation: SEC profiles over time of FapC incubated with (**B**) PGG, (**C**) oleuropein, and (**D**) without inhibitor.

**Figure 8 biomolecules-09-00659-f008:**
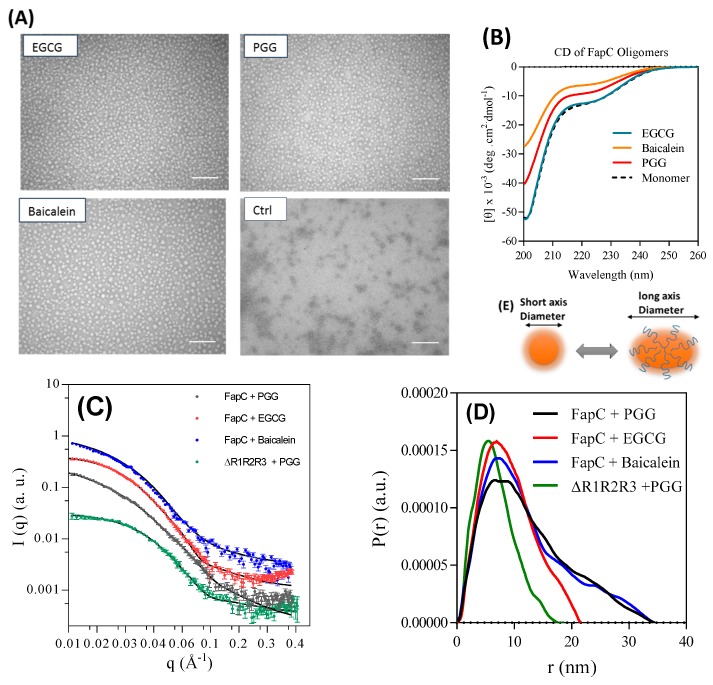
Structural analysis of off-pathway oligomers purified by SEC. (**A**) TEM image of oligomers show large spherical oligomers in the presence of inhibitors, in contrast to the small oligomers formed in their absence. Scale bars are 200 nm; (**B**) far-ultra violet (UV) CD spectra of monomer and SEC-purified oligomers both show largely random coil; (**C**) small-angle X-ray scattering (SAXS) data (scattering intensity *I (q)* versus length of scattering vector (q)) for purified oligomers all show the same overall shape; (**D**) pair distance distribution functions for the oligomers from panel c; (**E**) schematic representation of a possible monomer arrangement in the oligomer.

**Figure 9 biomolecules-09-00659-f009:**
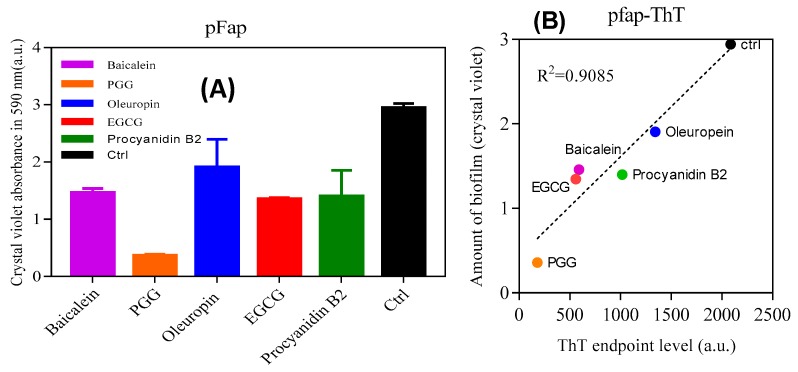
Effect of polyphenols on *Pseudomonas* biofilm. (**A**) The amount of biofilm formed in the presence of polyphenols (100 µM) by *Pseudomonas* overexpressing the *fap UK4* operon; (**B**) correlation between the amount of biofilm and endpoints level of ThT fluorescence for FapC protein incubated with and without polyphenols; (**C**) confocal microscopy images of pFap and wt incubated with and without 100 µM PGG and EGCG. The bacteria were stained with live/dead stain: Red and green denote dead and live bacteria, respectively, all images were taken with 63× oil objective, scale bars are 10 µm.

**Table 1 biomolecules-09-00659-t001:** SAXS data for monomers and SEC-purified oligomers of FapC.

Samples	Mw (kDa) *^a^*	*N* _agg_ *^b^*	R_g, shell_ (nm) *^c^*	R_g, total_ (nm) *^d^*	*f* _Shell_ *^e^*	Diameter (Short Axis) (nm)	Diameter (Long Axis) (nm)
Monomers (freshly desalted)							
FapC-TCEP	34.7 ± 1.0	1.4 ± 0.1		5.9 ± 0.1			
FapC	109 ± 3	4.4 ± 0.2		10.1 ± 0.2			
∆R1R2R3	38.9 ± 1.4	2.7 ± 0.1		7.0 ± 0.1			
∆R1R2R3-TCEP	26.1 ± 0.9	1.8 ± 0.1		5.9 ± 0.2			
Oligomer samples purified by SEC							
∆R1R2R3-PGG	60 ± 1	4.7 ± 0.1	1.1 ± 0.1	5.3 ± 0.2	0.27	19.5 ± 0.7	24.2 ± 0.7
FapC-PGG	190 ± 1	7.6 ± 0.2	1.7 ± 0.1	10.1 ± 0.2	0.31	27.2 ± 0.5	52.3 ± 0.5
FapC-EGCG	168 ± 1	6.7 ± 0.1	1.8 ± 0.1	7.2 ± 0.1	0.33	25.6 ± 0.3	32.4 ± 0.3
FapC-Baicalein	168 ± 1	6.7 ± 0.1	1.8 ± 0.1	10.0 ± 0.2	0.36	25.8 ± 0.4	48.1 ± 0.4

Notes: *^a^* Mw: Molecular weight determined by IFT. *^b^* N_agg_: Number of monomers in the oligomer determined by IFT. *^c^* R_g_: The R_g_ of the chains that are flexible in the shell. *^d^* R_g_: Determined with IFT for monomers and oligomers. *^e^ f*_shell_: Mass fraction of protein in the shell.

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
