# Peer review of "Plant Polyphenols Inhibit Functional Amyloid and Biofilm Formation in Pseudomonas Strains by Directing Monomers to Off-Pathway Oligomers"

_biomolecules, 2019, doi:10.3390/biom9110659_

Round 1

Reviewer 1 Report

In this work authors evaluated, utilizing an array of different techniques, the inhibitory effect of different polyphenols on the fibrillation of FapC, the major component of amyloid structure in the biofilm of many different Pseudomonas strains.

On the frame of this study it has been suggested that the evaluated polyphenols develop interactions with the linker region and the most amyloidogenic parts of the monomer FapC, with the most potent being PGG. As authors indicate their work could assist in utilizing natural products in controlling biofilm-related infections.

Below are some points that the authors should take into account to refine their work.

Although the work accumulates in a logical manner several biophysical techniques the authors have to present in a more rational way the reason that the specific 5 polyphenols have been selected. i.e. there is not a rational design strategy that a diverse scaffold has been used and also it is not explained the basis of PGG structural superiority with respect to the other polyphenols. A nice input in the work could be gallic acid that is a subscaffold of PGG and EGCG. The integration of gallic acid in this work could refine this work. It is expected to be the active component of PGG. Authors have to report in more detail the material and methods. i.e. it is not clear the conditions/solvents/stock solutions that polyphenols have been prepared. Some of the used polyphenols used are insoluble in water. Have the authors made the master stock in DMSO? If this is the case has DMSO been used as control. These are not clear. The stability (medium that the compounds have been tested) of the evaluated polyphenols has not been indicated i.e. PGG, EGCG, oleuropein contain labile ester bonds that can be rapidly cleaved in pH7.5 in some hours. Thus it is not clear if the observed biophysical results are the impact of the parent compounds or of the hydrolysed products.

Author Response

Reviewer 1:

In this work authors evaluated, utilizing an array of different techniques, the inhibitory effect of different polyphenols on the fibrillation of FapC, the major component of amyloid structure in the biofilm of many different Pseudomonas strains.

On the frame of this study it has been suggested that the evaluated polyphenols develop interactions with the linker region and the most amyloidogenic parts of the monomer FapC, with the most potent being PGG. As authors indicate their work could assist in utilizing natural products in controlling biofilm-related infections.

Below are some points that the authors should take into account to refine their work.

Although the work accumulates in a logical manner several biophysical techniques the authors have to present in a more rational way the reason that the specific 5 polyphenols have been selected. i.e. there is not a rational design strategy that a diverse scaffold has been used and also it is not explained the basis of PGG structural superiority with respect to the other polyphenols. A nice input in the work could be gallic acid that is a subscaffold of PGG and EGCG. The integration of gallic acid in this work could refine this work. It is expected to be the active component of PGG.

Response: We appreciate the opportunity to expand our investigations and put our selection on an even firmer basis. Based on the reviewer’s suggestions, we have now carried out additional Thioflavin T-based assays using 5 additional e polyphenols with similar subscaffolds. Gratifyingly, our choice of five compounds featured in our initial submission turned out to be vindicated, since the additional compounds all performed worse.

In the Introduction we write that “To investigate this, we have compared the activities of a number of different polyphenols against the fibrillation of FapC wildtype (wt) and truncations thereof. Starting with a broad range of polyphenols, we select EGCG, PGG, baicalein, oleuropein and procyanidin B2 as the most promising candidates.”

In Results we now start by writing “As an initial step, we compare the ability of ten different polyphenols to inhibit FapC fibrillation. These ten compounds are PGG, EGCG, oleuropein, baicalein and procyanidin-B2, verbascoside, 3-hydroxytyrosol, epicatechin, catechin and gallic acid (Fig. S1). All ten compounds contain multiple vicinal hydroxyl groups, ranging from single aromatic rings (gallic acid and 3-hydroxy-terosol) to complex systems linked through a glucose ring (PGG); furthermore, gallic acid and catechin/epicatechin are subscaffolds of EGCG, procyanidin and PGG. We incubated 30 µM wt FapC monomer with 10-200 µM polyphenol (Figure S1), monitoring fibril formation with the amyloid-binding dye Thioflavin T (ThT, representative runs in Figure 1BC). All compounds decrease ThT end-point emission in a dose-dependent fashion (Figure 1D), with PGG showing the strongest effect, followed by baicalein, EGCG, procyanidin B2 and oleuropein. The remaining 5 compounds showed significantly weaker effects, indicating that a reduction in compound size below a certain level of complexity leads to a loss of activity. Accordingly we proceed with the 5 best polyphenols.”

Finally, in the Discussion we add that “Prior to that, we established that smaller compounds such as gallic acid, catechin, epicatechin and 3-hydroxytyrosol, which represent subscaffolds of the larger polyphenols, performed poorly compared to the more complex polyphenols. This indicates cooperativity effects from combining several functional groups close in space.”

Authors have to report in more detail the material and methods. i.e. it is not clear the conditions/solvents/stock solutions that polyphenols have been prepared. Some of the used polyphenols used are insoluble in water. Have the authors made the master stock in DMSO? If this is the case has DMSO been used as control. These are not clear.

Response: Very salient point. We have dissolved the polyphenols first in DMSO to make a stock with high concentration (100 mM) and then diluted it. To clarify this we now add “(from stocks of 100 mM in DMSO)” in Materials and Methods. We also provide raw data in Fig. 1A to demonstrate the lack of effect of DMSO. In the same place we have therefore added “Note that the low residual amounts of DMSO from the stock solution did not affect fibrillation kinetics (Fig. 1A).”

The stability (medium that the compounds have been tested) of the evaluated polyphenols has not been indicated i.e. PGG, EGCG, oleuropein contain labile ester bonds that can be rapidly cleaved in pH7.5 in some hours. Thus, it is not clear if the observed biophysical results are the impact of the parent compounds or of the hydrolysed products.

We have analyzed the stability of polyphenols using reverse-phase HPLC. While there is a reduction in the stability over time, the compounds are largely intact over the first two critical hours. Furthermore, the NBT-assay (Figure 6B) indicates that EGCG reacts with monomers and oligomers mostly within the first incubation hour of time.

In the Materials and Methods we add the following description:

Reverse Phase High-performance liquid chromatography (RP-HPLC): 100 µM of polyphenol in 50 mM Tris buffer pH 7.5 was incubated on an Eppendorf thermoshaker (TS-100, BioSan, Latvia) at 37 ° at 300 rpm. After 0, 1, 2, 4, 8, 16 and 24 h of incubation, 20 µl of sample was injected onto a C18 column (RP-C18 Luna column, 4.6 mm id×250 mm, Phenomenex, UK) and elution was monitored by absorption at 270 nm. Material was eluted using a mobile phase of 30:70 v/v acetonitrile: water at a flow rate of 0.5 ml/min for all polyphenols except for the more hydrophobic compound baicalein where the mobile phase consisted of 80:20 v/v acetonitrile: water [1].“

In the Results section we add: “We used RP-HPLC to assess compound stability over 24 h at 37oC pH 7 (Fig. S2). While the population of the starting compound declined by 40-70% over 24 h, the compounds were largely stable (2-25% loss) over the first 2 h, corresponding to the lag phase of FapC fibrillation. Thus the compounds are essentially intact during this critical time period where fibrillation would otherwise start to occur. Furthermore, the compounds form stable adducts with FapC within the first hour of incubation (see below).”

Reviewer 2 Report

This submission screened the effects of polyphenols including epigallocatechin gallate (EGCG), penta-O-galloyl-β-D-glucose (PGG), baicalein, oleuropein and procyanidin B2 on the proteins to β-sheet rich amyloid fibrils extracted from the extracellular matrix of bacterial biofilm. I like to give the following comments.

As the amyloid, similarity of the protein FapC with that in human brain needs to introduce in clear. EGCG and PGG as the best inhibitors of FapC. Is it consistent with the animal studies? Limitation(s) for understanding of how polyphenols affect fibrillation of amyloidogenic proteins in this way must conduct in clear. It seems better to develop EGCG for treatment of aeroginosa biofilm. Role of antioxidant-like effect of EGCG or PGG needs to concern in addition. Final target of the present study remained unclear.

Author Response

This submission screened the effects of polyphenols including epigallocatechin gallate (EGCG), penta-O-galloyl-β-D-glucose (PGG), baicalein, oleuropein and procyanidin B2 on the proteins to β-sheet rich amyloid fibrils extracted from the extracellular matrix of bacterial biofilm. I like to give the following comments.

As the amyloid, similarity of the protein FapC with that in human brain needs to introduce in clear.

Response: In the Introduction we note that “Unlike misfolded proteins involved in amyloid-related diseases like Parkinson’s and Alzheimer’s [2,3] , expression and release of these functional amyloidogenic proteins is meticulously controlled in biofilm formation and leads to useful biological properties…”. To emphasize the difference to pathological amyloid, we now add the following at appropriate points in the Introduction:

“Unlike pathological amyloid in e.g. neurodegenerative diseases, production and secretion of functional amyloid is carefully regulated by a number of helper proteins.” and “Control of functional amyloid formation is also seen at the level of the protein sequence.”

EGCG and PGG as the best inhibitors of FapC. Is it consistent with the animal studies?

Response: To expand on the role of polyphenols in in vivo studies, we add the following to the Discussion: After “treatment of P.aeroginosa biofilm by EGCG increased its susceptibility to antibiotics and decreased its stiffness [2]” we add “while the polyphenol tannic acid inhibits the ability of S. aureus to colonize surfaces [54]”.  We also add “Use of polyphenols against biofilm is also encouraged by the observations that e.g. EGCG [3] and oleuropein [4] do not show toxicity issues in animal models and EGCG has even progressed to clinical trials for use against Alzheimer’s Disease [5].”

Limitation(s) for understanding of how polyphenols affect fibrillation of amyloidogenic proteins in this way must conduct in clear.

Response: We assume the reviewer is referring to possible limitations in the use of polyphenols to prevent amyloid formation. We have used a large number of different approaches to establish how the polyphenols affect fibrillation of FapC, including effects on lag time, seeding, extent of aggregation and accumulation of other aggregated species. Importantly, we identify potential binding sites on the protein which can interact with polyphenols as part of the inhibitory mechanism. Thus we believe that we have in fact provided as detailed an analysis of the process as is possible to obtain from existing available techniques, finally linking it to their ability to affect biofilm formation. Further steps might involve molecular dynamics simulations of the effects, but this is unfeasible at present due to the possible involvement of chemical reactions in the process. However, we acknowledge that there could be some challenges due to their relative chemical instability. Accordingly we have added the following in the Discussion: “While our polyphenols effectively inhibit biofilm formation in vivo, their chemical stability remains an issue, given that the concentration of the original compound declines significantly over a 24-h period. However, interactions with existing amyloid components (particularly chemical modifications) occur within the first hour (as detected by our NBT assay). Given that fibrillation of our functional proteins shows a lag time of several hours, there is clearly ample time for the polyphenols to interact with their protein targets before fibrillation takes off in earnest. In practice, regular supplies of fresh polyphenols (e.g. on a daily basis) would likely to sufficient to ensure full efficacy against unwanted bacterial protein fibrillation.”

It seems better to develop EGCG for treatment of aeroginosa biofilm.

Response: We agree that this would be a major advantage. We have in fact addressed the specific use of EGCG against P. aeruginosa biofilm in a previous publication (Stenvang et al., JBC 2016, ref. 26). We refer to this work and its conclusions on several occasions, both in the Introduction, Results and Discussion. Note that analysis of P. aeruginosa biofilm is more complicated than other Pseudomonas biofilm due to excessive alginate formation, leading to mucoid biofilm which is more challenging to analyze. Accordingly in Discussion we add after reference to P. aeruginosa: “(though high levels of alginate production complicated analysis of biofilm production in this species).”

Role of antioxidant-like effect of EGCG or PGG needs to concern in addition.

Response: In the Discussion, we add the following: “Furthermore, it should be noted that their chemical reactivity is connected to their anti-oxidant potential, most likely through resonance stabilization of oxidation-induced free radical states [6, 7]. PGG scavenges reactive oxygen species such as superoxide (O2) and H2O2, enhancing resistance of neuronal cells to hydrogen peroxide [8, 9].”

Final target of the present study remained unclear.

Response: To emphasize the focus of our study, in the Abstract we now add “Here we elucidate how polyphenols modify the self-assembly of functional amyloid, with particular focus on…” as well as adding “We correlate our biophysical observations to biological impact by demonstrating that” before “the extent of amyloid inhibition by the different inhibitors correlated with their ability to reduce biofilm”. Thus it should hopefully be quite clear that we wish to elucidate mechanisms of aggregation inhibition and their impact in vivo.

Reviewer 3 Report

Plant polyphenols are  important parts of human food. Therefore, the bio-activities are always timely topics of investigations. The authors submitted an original manuscript, which deals with an apparent inhibition of amyloid  formation of Pseudomonas spp. by selected plant polyphenols. Pseudomonas spp. is a gram negative bacteria, which occur ubiquitously and cause often opportune or dangerous nosocomial infections. In this context, the results presented in this manuscript are interesting.

The authors used adequate methods. The data support the results, which are illustrated by many graphs or figures. Additionally, I appreciate especially the illustrations in the supplementary materials. Moreover, the results are properly discussed. Therefore, I recommend the manuscript for publication. 

Author Response

We are grateful for the appreciative comments by Reviewer 3.

Round 2

Reviewer 1 Report

The authors have responded properly on the raised comments.